# GENERATIVE ADVERSARIAL INTERPOLATIVE AUTOENCODING: ADVERSARIAL TRAINING ON LATENT SPACE INTERPOLATIONS ENCOURAGES CONVEX LATENT DISTRIBUTIONS

## ABSTRACT

We present a neural network architecture based upon the Autoencoder (AE) and Generative Adversarial Network (GAN) that promotes a convex latent distribution by training adversarially on latent space interpolations. By using an AE as both the generator and discriminator of a GAN, we pass a pixel-wise error function across the discriminator, yielding an AE which produces sharp samples that match both high- and low-level features of the original images. Samples generated from interpolations between data in latent space remain within the distribution of real data as trained by the discriminator, and therefore preserve realistic resemblances to the network inputs.

## 1 INTRODUCTION

Generative modeling has the potential to become an important tool for exploring the parallels between perceptual, physical, and physiological representations in fields such as psychology, linguistics, and neuroscience (e.g. Sainburg et al. 2018; Thielk et al. 2018; Zuidema et al. 2018). The ability to infer abstract and low-dimensional representations of data and to sample from these distributions allows one to quantitatively explore and vary complex stimuli in ways which typically require hand-designed feature tuning, for example varying formant frequencies of vowel phonemes, or the fundamental frequency of syllables of birdsong.

Several classes of unsupervised neural networks such as the Autoencoder (AE; Hinton & Salakhutdinov 2006; Kingma & Welling 2013), Generative Adversarial Network (GAN; Goodfellow et al. 2014), autoregressive models (Hochreiter & Schmidhuber, 1997; Graves, 2013; Oord et al., 2016; Van Den Oord et al., 2016), and flow-based generative models (Kingma & Dhariwal, 2018; Dinh et al., 2014; Kingma et al., 2016; Dinh et al., 2016) are at present popularly used for learning latent representations that can be used to generate novel data samples. Unsupervised neural network approaches are ideal for data generation and exploration because they do not rely on hand-engineered features and thus can be applied to many different types of data. However, unsupervised neural networks often lack constraints that can be useful or important for psychophysical experimentation, such as pairwise relationships between data in neural network projections, or how well morphs between stimuli fit into the true data distribution.

We propose a novel AE that hybridizes features of an AE and a GAN. Our network is trained explicitly to control for the structure of latent representations and promotes convexity in latent space by adversarially constraining interpolations between data samples in latent space to produce realistic samples[1].

---

[1] Pairwise interpolations in between latent samples may only cover a subset of the convex hull of the latent distribution, as described in Figure 1.

## 1.1 BACKGROUND ON GENERATIVE ADVERSARIAL NETWORKS (GANS) AND AUTOENCODERS (AES)

An AE is a form of neural network which takes as input $x_i$ (e.g. an image), and is trained to generate a reproduction of the input[2] $G(x_i)$, by minimizing some error function between the input $x_i$ and output $G(x_i)$ (e.g. pixel-wise error). This translation is usually performed after compressing the representation $x_i$ into a low-dimensional representation $z_i$. This low-dimensional representation is called a latent representation, and the layer corresponding to $z_i$ in the neural network is often called the latent layer. The first half of the network, which translates from $x_i$ to $z_i$, is called the encoder; the second half of the network, which translates from $z_i$ to $x_i$, is called the decoder. The combination of these two networks make the AE capable of both dimensionality reduction ($X \rightarrow Z$); Hinton & Salakhutdinov 2006), and generativity ($Z \rightarrow X$). Importantly, AE architectures are generative[3], however they are not generative *models* (Bishop, 2006) because they do not model the joint probability of the observable and target variables $P(X, Z)$. Variants such as the Variational Autoencoder (VAEs; Kingma & Welling 2013), which model $P(X, Z)$ are generative models. AE latent spaces, therefore, cannot be sampled probabilistically, without modeling the joint probability as in VAEs. The AE architecture that we propose here does not model the joint probability of $X$ and $Z$ and thus is not a generative model, although the latent space of our network could be modeled probabilistically (e.g. with a VAE).

GAN architectures are comprised of two networks, a generator, and a discriminator. The generator takes as input a latent sample, $z_i$, drawn randomly from a distribution (e.g. uniform or normal), and is trained to produce a sample $G_d(z_i)$ in the data domain $X$. The discriminator takes as input both $x_i$ and $G_d(z_i)$, and is trained to differentiate between real $x_i$, and generated $G_d(z_i)$ samples, typically by outputting either a 0 or 1 in a single-neuron output layer. The generator is trained to oppose the discriminator by 'tricking' it into categorizing $G_d(z)$ samples as $x$ samples. Intuitively, this results in the generator producing $G_d(Z)$ samples indistinguishable (at least to the discriminator) from those drawn from the distribution $x$. Thus the discriminator acts as a 'critic' of the samples produced by a generator that is attempting to reproduce the distribution $x$. Because GANs sample directly from a predefined latent distribution, GANs are *generative models*, explicitly representing the joint probability, $P(X, Z)$.

One common use for both GANs and AEs has been exploiting the semantically rich low-dimensional manifold, $Z$, on which data are either projected onto or sampled from (White, 2016; Hinton & Salakhutdinov, 2006). Operations performed in $Z$ carry rich semantic features of data, and interpolations between points in $Z$ produce semantically smooth interpolations in the original data space $X$ (e.g. Radford et al. 2015; White 2016). However, samples generated by latent representations of both AEs and GANs are limited by the constraints provided by the algorithm. A significant amount of work has been done over the past several years in developing variants of AEs and GANs which add additional constraints and functionality to GAN and AE architectures, for example improving stability of GANs (e.g. Berthelot et al. 2017; Radford et al. 2015; Salimans et al. 2016), disentangling latent representations (e.g. Higgins et al. 2016; Chen et al. 2016; Bouchacourt et al. 2017), adding generative capacity to AEs (e.g. Kingma & Welling 2013; Kingma et al. 2016; Makhzani et al. 2015), and adding bidirectional inference to GANs (e.g. Larsen et al. 2015; Mescheder et al. 2017; Berthelot et al. 2017; Dumoulin et al. 2016; Ulyanov et al. 2017; Makhzani 2018).

In this work, we describe several limitations of GANs and Autoencoders, specifically as they relate to stimuli generation for psychophysical research, and propose a novel architecture, GAIA, that utilizes aspects of both the AE and GAN to negate shortcomings of each architecture independently. Our method provides a novel approach to increasing the stability of network training, increasing the convexity of latent space representations, preserving of high-dimensional structure in latent space representations, and bidirectionality from $X \rightarrow Z$ and $Z \rightarrow X$.

## 1.2 CONVEXITY OF LATENT SPACE

Generative latent-spaces enable the powerful capacity for smooth interpolations between real-world signals in a high-dimensional space. Linear interpolations in a low-dimensional latent space often

---

[2]We denote $G(x_i)$ as being equivalent to $G_d(G_e(x_i))$, or $x_i$ being passed through the encoder and decoder of the generator, $G$

[3]having the power or function of generating, originating, producing, or reproducing (Webster, 2018)

produce comprehensible representations when projected back into high-dimensional space (e.g. Engel et al. 2017; Dosovitskiy et al. 2015). In the latent spaces of many network architectures such as AEs, however, linear interpolations are not necessarily justified, because the space between endpoints on an interpolation in $Z$ is not explicitly trained to fall within the data distribution when translated back into $X$.

A convex set of points is defined as a set in which the line-segment connecting any pair of points will fall within the rest of the set (Klee, 1971). For example, in Figure 1A, the purple distribution represents data projected into a two-dimensional latent space, and the surrounding whitespace represents regions of latent space that do not correspond to the data distribution. This distribution would be non-convex because a line connecting two points in the distribution (e.g. the black points in Figure 1A) could contain points outside the data distribution (the red region). In an AE, if the red region of the interpolation were sampled, projections back into the high-dimensional space may not necessarily correspond to realistic exemplars of $x$, because that region of $Z$ does not belong to the true data distribution.

One approach to overcoming non-convexity in a latent space is to force the latent representation of the dataset into a pre-defined distribution (e.g. a normal distribution), as is performed by VAEs. By constraining the latent space of a VAE to fit a normal distribution, the latent space representations are encouraged to belong to a convex set. This method, however, pre-imposes a distribution in latent space that may be a suboptimal representation of the high-dimensional dataset. Standard GAN latent distributions are sampled directly, similarly allowing arbitrary convex distributions to be explicitly chosen for latent spaces. In both cases, hard-coding the distributional structure of the latent space may not respect the high-dimensional structure of the original distribution.

### 1.3 PIXEL-WISE ERROR AND BIDIRECTIONALITY

AEs that perform dimensionality reduction (in particular VAEs) can produce blurry images due to their pixel-wise loss functions (Goodfellow et al., 2014; Larsen et al., 2015), which minimize loss by smoothing the sharp contrasts (e.g. edges) present in real data. GANs do not suffer from this blurring problem, because they are not trained to reproduce input data. Instead, GANs are trained to generate data that could plausibly belong to the true distribution $X$. Thus, smoothing over uncertainty tends to be discouraged by the discriminator because it can use smoothed edges as a distinguishing feature between data sampled from $X$ and $G(X)$.

Producing data that fits into the distribution of $x$, rather than reproducing individual instances of $x_i$ comes at a cost, however. While AEs learn both the translation from $X$ to $Z$ and $Z$ to $X$, GANs only learn the latter ($Z \rightarrow X$). In other words, the pixel-wise loss function of the AE produces smoothed data but is bidirectional, while the discriminator-based loss function of the GAN produces sharp images and is unidirectional.

## 2 GENERATIVE ADVERSARIAL INTERPOLATIVE AUTOENCODING (GAIA)

Our model, GAIA (Figure 1 left), is bidirectional but is trained on both a GAN loss function and a pixel-wise loss function, where the pixel-wise loss function is passed across the discriminator of the GAN to ensure that features such as blurriness are discriminated against. In full, GAIA is trained as a GAN in which both the generator and the discriminator are AEs. The discriminator is trained to minimize the pixel-wise loss ($\ell_1$) between real data ($x_i$) and their AE reproduction in the discriminator ($D(x_i)$) while maximizing the AE loss between samples generated ($G(x_i)$) by the generator and their reproduction in the discriminator ($D(G(x_i))$):

$$\|x_i - D(x_i)\|_1 - \|x_i - D(G(x_i))\|_1$$

The generator is trained on the inverse, to minimize the pixel-wise loss between input ($x_i$) and output ($D(G(x_i))$) such that the discriminator reproduces the generated samples to be as close to the original data as possible:

$$\|x_i - D(G(x_i))\|_1$$

Using an AE as a generator has been previously been used in the VAE-GAN (Larsen et al., 2015), and decreases blurring from the pixel-wise loss in AEs at the expense of exact-reproduction of data. Similarly, using an AE as a discriminator has been previously used in BEGAN (Berthelot

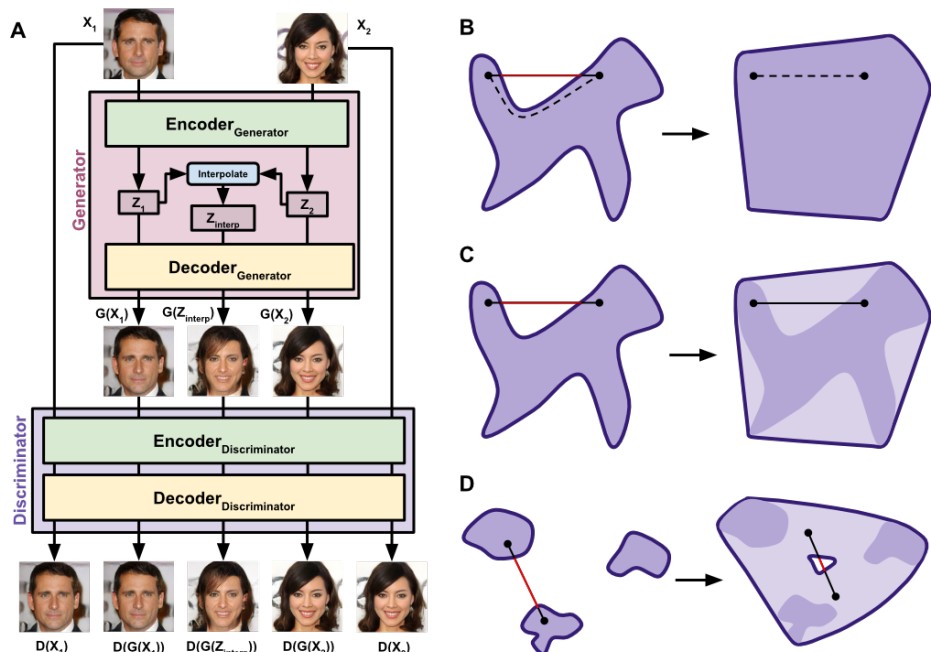

Figure 1: **(A)** GAIA network architecture. **(B)** An example AE latent distribution (purple) which is non-convex. Black circles represent two projections from $X$ to $Z$. The solid line between them represents an interpolation, where the red region contains points in the interpolation that do not correspond to the true data distribution. The dashed line is an interpolation which passes through only the true data distribution. One hypothesis is that training will warp the distribution on the left to produce linearly interpolations within the distribution. **(C)** An alternative hypothesis is that by training the network upon interpolations, samples in the interpolated regions (lighter purple) will be trained to generate data similar to the $x$, without manipulating the distribution of $z$. **(D)** Pairwise interpolations between samples of $x$ in GAIA will not necessarily make the latent distribution convex, because two point interpolations in $Z$ do not reach all of the points in between the interpolated data $z_{int.}$.

et al., 2017), which improves stability in GANs but remains unidirectional[4]. In GAIA, we combine these two architectures, allowing the generator to be trained on a pixel-wise loss that is passed across the discriminator, explicitly reproducing data as in an AE, while producing sharper samples characteristic of a GAN.

In addition, linear interpolations are taken between the latent-space representations:

$$\beta \leftarrow \mathcal{N}(\mu, \sigma^2)$$

$$z_{int.} = z_{gen_i}\beta + z_{gen_j}(1 - \beta)$$

Where interpolations are Euclidean interpolations between pairs of points in $Z$, sampled from a 1-dimensional Gaussian distribution[5] centered around the midpoint between $z_{gen_i}$ and $z_{gen_j}$. The midpoints are then passed through the decoder of the discriminator, which are treated as generated samples by the GAN loss function:

$$\|G_d(z_{int.}) - D(G_d(z_{int.}))\|_1$$

The discriminator is trained to maximize this loss, and the generator is trained to minimize this loss.

---

[4]Although it is possible to find the regions of $Z$ most closely corresponding to $x_i$

[5]$\sigma = 0.25$. We sample along the midpoint using a Gaussian rather than uniformly because we found that interpolations near to original samples required less training than interpolations to produce realistic interpolations.

In full, the loss of the discriminator, as in BEGAN, is to minimize pixel-wise loss of real data, and maximize pixel-wise loss of generated data (including interpolations):

$$L_{Disc} = \|x_i - D(x_i)\|_1 - \\ \|x_i - D(G(x_i))\|_1 - \\ \|G_d(z_{int.}) - D(G_d(z_{int.}))\|_1$$

The loss of the generator is to minimize the error across the descriminator for the input data in the generator ($D(G(x_i))$), along with the minimizing the error of the interpolations generated by the generator ($D(G_d(z_{int.}))$).

$$L_{Gen} = \|x_i - D(G(x_i))\|_1 + \\ \|G_d(z_{int.}) - D(G_d(z_{int.}))\|_1$$

In summary, the generator and discriminator are both AEs. As a result, reconstructions of $x$ have the potential to resemble the input data ($G(x)$) at a pixel level, a feature non-existent in other GAN based inference methods (Figure 5). We also train the network on interpolations in the generator, to explicitly train the generator to produce interpolations ($G_d(z_{int.})$) which deceive the discriminator and are closer to the distribution in $X$ than interpolations from an unconstrained AE.

## 2.1 PRESERVATION OF LOCAL-STRUCTURE IN HIGH-DIMENSIONAL DATA

VAEs and GANs force the latent distribution, $z$, into a pre-defined distribution, for example, a Gaussian or uniform distribution. This approach presents a number of advantages, such as ensuring latent space convexity and thus being better able to sample from the distribution. However, these benefits are gained at the loss of respecting the structure of the distribution of the original high dimensional dataset, $x$. Preserving high-dimensional structure in low dimensional embeddings is often the goal of dimensionality reduction, one of the functions of an autoencoder (Hinton & Salakhutdinov, 2006). To better respect the original high-dimensional structure of the dataset, we impose a regularization between the latent space representations of the data ($z$) and the original high dimensional dataset ($x$), motivated by Multidimensional Scaling (Kruskal, 1964).

For each minibatch presented to the network, we compute a loss for the distance between the log of the pairwise Euclidean distances of samples in $X$ and $Z$ space:

$$L_{dist}(x, z) = \frac{1}{B} \sum_{i,j}^{B} \left[ log_2 \left( 1 + \frac{(x_i - x_j)^2}{\frac{1}{B} \sum_{i,j} (x_i - x_j)^2} \right) - log_2 \left( 1 + \frac{(z_i - z_j)^2}{\frac{1}{B} \sum_{i,j} (z_i - z_j)^2} \right) \right]^2$$

We then apply this error term to the generator to encourage the pairwise distances of the minibatch in latent space to be similar to the pairwise distances of the minibatch in the original high-dimensional space.

## 3 EXPERIMENTS

Here we apply out network architecture to two datasets: *(1)* five 2D distributions from Scikit-learn (Pedregosa et al., 2011) which allows us to visualize and quantify the behavior of GAIA in a low-dimensional space (Figure 2), and *(2)* the CELEBA-HQ dataset (Liu et al., 2015; Karras et al., 2017) which allows us to test the performance of GAIA on a more complex high dimensional dataset (Figure 3).

## 3.1 2D DATASETS

We compared the performance of AE, VAE, and GAIA networks with the same architecture and training parameters on five 2D distributions from Scikit Learn (Pedregosa et al., 2011). We also compared the GAIA network with and without the distance loss term ($L_{dist} = 0$). We computed the learned latent representations ($z$) as well as reconstructions ($G_d(z)$) from of each of the networks (Figures 2, 6), and compared the these spaces on a number of metrics (Table 1).

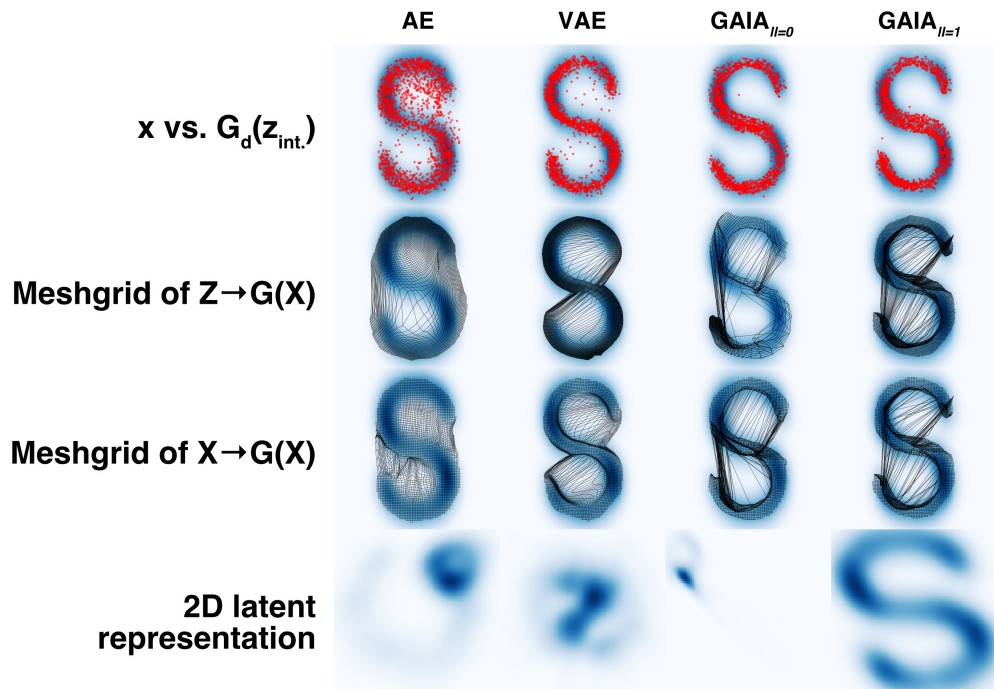

Figure 2: GAIA vs. AE and VAE on reconstruction on the *S* dataset. GAIA is shown both with the local-structure (ll=1) loss and without (ll=0). Results for the other four 2D datasets are shown in Figure 6. (Top) Interpolations in latent space $G_d(z_{int.})$ in red, plotted over true distribution plotted as blue heat-map. (Second row) A mesh-grid showing translation from a uniform sampling of the convex hull of $z$ reconstructed as $G_d(Z)$. (Third row) A mesh-grid showing translation from the convex hull of $x$ reconstructed as $G(X)$. (Bottom) The probability distribution of representations for each network, showing that the $L_{dist}$ term promotes the preservation of structure from $X$ to $G(X)$.

Our most salient observation can be found in the mesh-grids in Figure 2, where a clear boundary exists in the warping of high- and low-probability data in GAIA, as opposed to an autoencoder without adversarial regulation. A similar warping of low-probability data is seen in the VAE, although a smoother warping is seen at the boundaries.

We quantitatively analyzed the results of Figure 2 in Table 1. We found that interpolations in GAIA ($G_d(z_{int.})$) are the most likely to fall into the distribution of $x$ (Figure 2 top; $log(\mathcal{L}(G_d(z_{int.})))$). We also found that the distributions of both network reconstructions and interpolations in $Z$ most highly match the input distribution ($x$) in the VAE network (measured by Kullback-Leibler divergence). This is likely due to the adversarial loss in GAIA. While VAEs are trained to match the distribution of $x$, GAIA's generator is trained to find regions of $X$ which are sufficiently high-enough probability that the discriminator will not discriminate against it. Finally, we found that pairwise Euclidean distances in $Z$ most highly resembled the original data distribution $x$ ($r(x, z)$) in the GAIA network when the $L_{dist}$ loss was imposed on the network. This leads us to conclude that GAIA can learn to map interpolations in latent-space onto the true data distribution in $X$ in a similar manner as a VAE, while still respecting the original structure of the data.

## 3.2 CELEBA-HQ

To observe the performance of our network on a more complex and high dimensional data, we use the CELEBA-HQ image dataset of aligned celebrity faces. We find that interpolations in $Z$ produce smooth realistic morphs in $X$ (Figure 3), and that complex features can be manipulated as linear vectors in the low-dimensional latent space of the network (Figure 4).

Table 1: Comparison of GAIA, VAE, and AE on 2D datasets.

| Model | $r(x,z)$ [*] | $log(\mathcal{L}(G_d(z_{int.})))$ [‡] | $log(\mathcal{L}(G(x)))$ | $D_{KL}(x \parallel G_d(z_{int.}))$ [†] | $D_{KL}(x \parallel G(x))$ |
|---|---|---|---|---|---|
| AE | 0.58 | 3207.39 | 3545.63 | 0.43 | -0.57 |
| VAE | 0.64 | 3574.11 | **3588.50** | **-0.05** | **-0.64** |
| GAIA$_{ll=0}$ | 0.38 | 3567.49 | 3564.27 | 0.19 | -0.36 |
| GAIA$_{ll=1}$ | **0.91** | **3593.84** | 3563.91 | 0.36 | -0.32 |

Results are intercepts from an OLS regression controlling for 2D dataset type, thus some values (such as KL divergence) can be negative.
[*]Pearson correlation [‡]Log-likelihood [†]Kullback-Leibler divergence

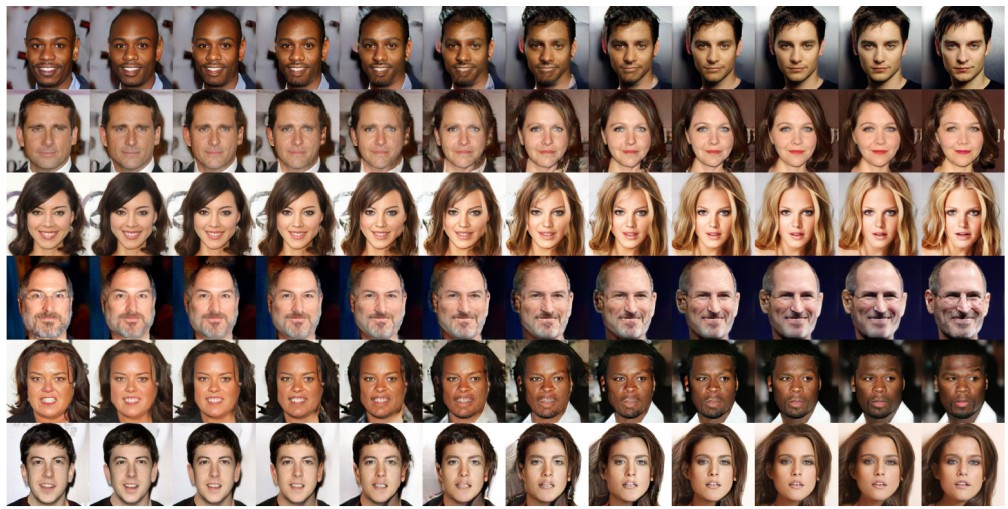

Figure 3: Interpolations between autoencoded images in our network. The farthest left and right columns correspond to the input images, and the middle columns correspond to a linear interpolation in latent-space.

### 3.2.1 FEATURE MANIPULATION

Feature manipulation using generative models typically fall into two domains: *(1)* fully unsupervised approaches, where feature vectors are extracted and applied after learning (e.g. Radford et al. 2015; Larsen et al. 2015; Kingma & Dhariwal 2018; White 2016), and *(2)* supervised and partially supervised approaches, where high-level feature information is used during learning (e.g. Choi et al. 2017; Isola et al. 2017; Li et al. 2016; Perarnau et al. 2016; Zhu et al. 2017).

We find that, similar to the latter group of models, high-level features correspond to linear vectors in GAIA's latent spaces (Figure 4). High-level feature representations are typically determined using the means of $Z$ representations of images containing features (e.g Radford et al. 2015; Larsen et al. 2015). The mean of the latent representations of the faces in the dataset (here CELEBA-HQ) containing an attribute ($z_{feat}$) and not containing that attribute ($z_{nofeat}$) is subtracted ($z_{feat} - z_{nofeat}$) to acquire a high-level feature vector. The feature vector is then added to, or subtracted from, the latent representation of individual faces ($z_i$), which is passed through the decoder of the generator, producing an image containing that high-level feature.

Similar to White (2016), we find that this approach is confounded by features being tangled together in the CELEBA-HQ dataset. For example, adding a latent vector to make images look *older* biases the image toward *male*, and making the image look more *young* biased the image toward *female*. This likely happens because the ratio of young males to older males is 0.28:1, whereas the ratio of young females to older females is much greater at 8.49:1 in the CELEBA-HQ dataset. As opposed to White (2016), who balance samples containing features in the training dataset, we use the coefficients of an ordinary least-squares regression trained to predict $z$ representations from feature attributes on the full dataset as feature vectors. We find that these features (Figure 7 bottom) are less intertwined than subtracting means alone (Figure 7 top).

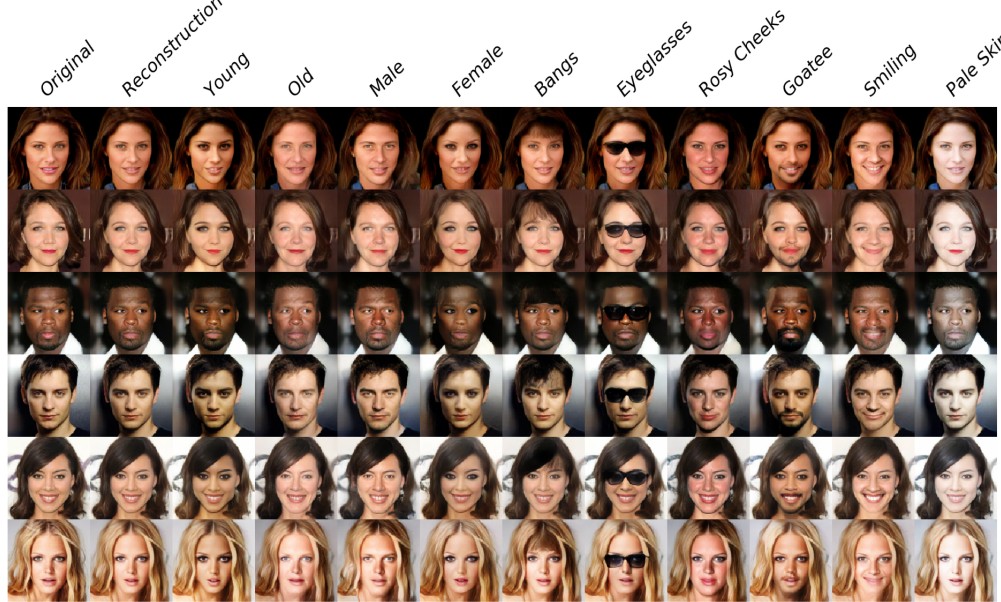

Figure 4: Attribute vectors added to $Z$ representation of different images from the CELEBA-HQ dataset.

### 3.3 RELATED WORK

This work builds primarily upon the GAN and AE. We used the AE as a discriminator motivated by Berthelot et al. (2017), and an AE as a generator motivated by Larsen et al. (2015), which in concert act as both an autoencoder and a GAN imparting bidirectionality on a GAN and imparting an adversarial loss on the autoencoder. A number of other adversarial network architectures (e.g. Larsen et al. 2015; Mescheder et al. 2017; Berthelot et al. 2017; Dumoulin et al. 2016; Ulyanov et al. 2017; Makhzani 2018) have been designed with a similar motivation in recent years. Our approach differs from these methods in that, by using an autoencoder as the discriminator, we are able to use a reconstruction loss which is passed across the discriminator, resulting in pixel-wise data reconstructions (Figure 5).

Similar motivations for better bidirectional inference-based methods have also been explored using flow-based generative models (Kingma & Dhariwal, 2018; Dinh et al., 2014; Kingma et al., 2016; Dinh et al., 2016), which do not rely on an adversarial loss. Due to their exact latent-variable inference (Kingma & Dhariwal, 2018), these architectures may also provide a useful direction for developing generative models to explore latent-spaces of data for generating datasets for psychophysical experiments.

In addition, the first revision of this work was published concurrently to ACAI network (Berthelot et al., 2018), which also uses an adversarial constraint on interpolations in the latent space of an autoencoder. Berthelot et al. find that adversarially constrained latent representations improve downstream tasks such as classification and clustering. At a high level, GAIA and ACAI networks perform the same functions, however, there are a few notable differences between the two networks. While ACAI uses an autoencoder as the discriminator of the adversarial network to improve pass the autoencoder error function across the discriminator, ACAI uses a traditional discriminator. As a result, the loss function is different between the two networks. Further comparisons are needed between the two architecture to compare network features such as training stability, reconstruction quality, latent feature representations, and downstream task performance.

## 4 CONCLUSION

We propose a novel GAN-AE hybrid in which both the generator and discriminator of the GAN are AEs. In this architecture, a pixel-wise loss can be passed across the discriminator producing autoencoded data without smoothed reconstructions. Further, using the adversarial loss of the GAN, we train the generator's AE explicitly on interpolations between samples projected into latent space, promoting a convex latent space representation. We find that in our 2D dataset examples, GAIA performs equivalently to a VAE in projecting interpolations in $Z$ onto the true data distribution in $X$, while respecting the original structure in $X$. We conclude that our method more explicitly lends itself to interpolations between complex signals using a neural network latent space, while still respecting the high-dimensional structure of the input data.

The proposed architecture still leaves much to be accomplished, and modifications of this architecture may prove to be more useful, for example utilizing different encoder strategies such as progressively growing layers (Karras et al., 2017), interpolating across the entire minibatch rather than two-point interpolations, modeling the joint probability of X and Z, and exploring other methods to train more explicitly on a convex latent space. Further explorations are also needed to understand how interpolative sampling effects the structure of the latent space of GAIA in higher dimensions.

Our network architecture furthers generative modeling by providing a novel solution to maintaining pixel-wise reconstruction over an adversarial architecture. Further, we take a step in the direction of convex latent space representations in a generative context. This architecture should prove useful both for current behavioral scientists interested in sampling from smooth and plausible stimuli spaces (e.g. Sainburg et al. 2018; Thielk et al. 2018; Zuidema et al. 2018), as well as providing motivation for future solutions to structured latent representations of data.

Our network was trained using Tensorflow, and our full model, code, and high-resolution images, along with videos of the model will be made available when de-anonymized.

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

## 5 APPENDIX

### 5.1 NETWORK ARCHITECTURE

In principle, any form of AE network can be used in GAIA. In the experiments shown in this paper, we used two different types of networks. For the 2D dataset examples, we use 6 fully connected layers per network with 256 units per layer, and a latent layer with two neurons. For the CELEBA-HQ dataset a modified version of network architecture advocated by Huang et al. (2018), which is comprised of a style and content AE using residual convolutional layers. Each layer of the decoder uses half as many filters as the encoder, and a linear latent layer is used in the encoder network but not the decoder network. The final number of latent neurons for the style and content networks are both 512 in the $128 \times 128$ pixel model shown here. The loss term for the pairwise-distance loss term is set at 2e-5. A Python/Tensorflow implementation of this network architecture is linked in the Conclusions section, and more details about the network architecture used are located in Huang et al. (2018).

### 5.2 INSTABILITY IN ADVERSARIAL NETWORKS

GANs are notoriously challenging to train, and refining techniques to balance and properly train GANs has been an area of active research since the conception of the GAN architecture (e.g. Berthelot et al. 2017; Salimans et al. 2016; Mescheder et al. 2017; Arjovsky et al. 2017). In traditional GANs, a balance needs to be found between training the generator and discriminator, otherwise one network will overpower the other and the generator will not learn a representation which fits the dataset. With GAIA, additional balances are required, such as between reproducing real images vs. discriminating against generated images, or balancing the generator of the network toward emphasizing autoencoding vs. producing high-quality latent-space interpolations.

We propose a novel, but simple, GAN balancing act which we find to be very effective. In our network, we balance the GAN's loss using a sigmoid centered at zero:

$$sigmoid(d) = \frac{1}{1 + e^{-d*b}}$$

In which $b$ is a hyper-parameter representing the slope of the sigmoid[6], and $d$ is the difference between the two values being balanced in the network. For example, the balance in the learning rate of the discriminator and generator is based upon the loss of the real and generated images:

$$\delta_{Disc} \leftarrow sigmoid(\|x_i - D(x_i)\|_1 - \|x_i - D(G(x_i))\|_1 + \|G_d(z_{int.}) - D(G_d(z_{int.}))\|_1/2)$$

The learning rate of the generator is then set as the inverse:

$$\delta_{Gen} \leftarrow 1 - \delta_{Disc}$$

This allows each network to catch up to the other network when it is performing worse. The same principles are then used to balance the different losses within the generator and discriminator, which can be found in Algorithm 1. This balancing act allows the part of the network performing more poorly to be emphasized in the training regimen, resulting in more balanced and stable training.

---

[6]kept at 20 for our networks

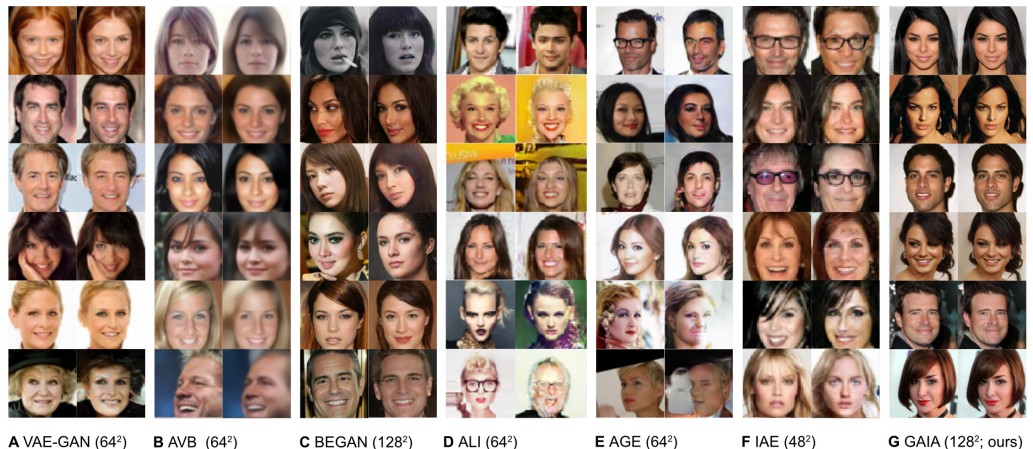

**A** VAE-GAN ($64^2$)  **B** AVB ($64^2$)  **C** BEGAN ($128^2$)  **D** ALI ($64^2$)  **E** AGE ($64^2$)  **F** IAE ($48^2$)  **G** GAIA ($128^2$; ours)

Figure 5: Data reconstructions from a subset of bidirectional GAN network architectures. Input images ($x$), and network reconstruction images ($G(x)$) are shown side by side, with inputs on the left. **(A)** Larsen et al. (2015) **(B)** Mescheder et al. (2017) **(C)** Berthelot et al. (2017) **(D)** Dumoulin et al. (2016) **(E)** Ulyanov et al. (2017) **(F)** Makhzani (2018) **(G)** Our method. Because most bidirectional GANs are either not trained on pixel-wise reconstruction, or do not pass pixel-wise reconstruction across the discriminator, reconstruction is either smoothed out, or exhibits features not present in the original data. Note that these methods all use different architectures, image data, and image resolutions, and therefore should not be compared for signal reconstruction quality.

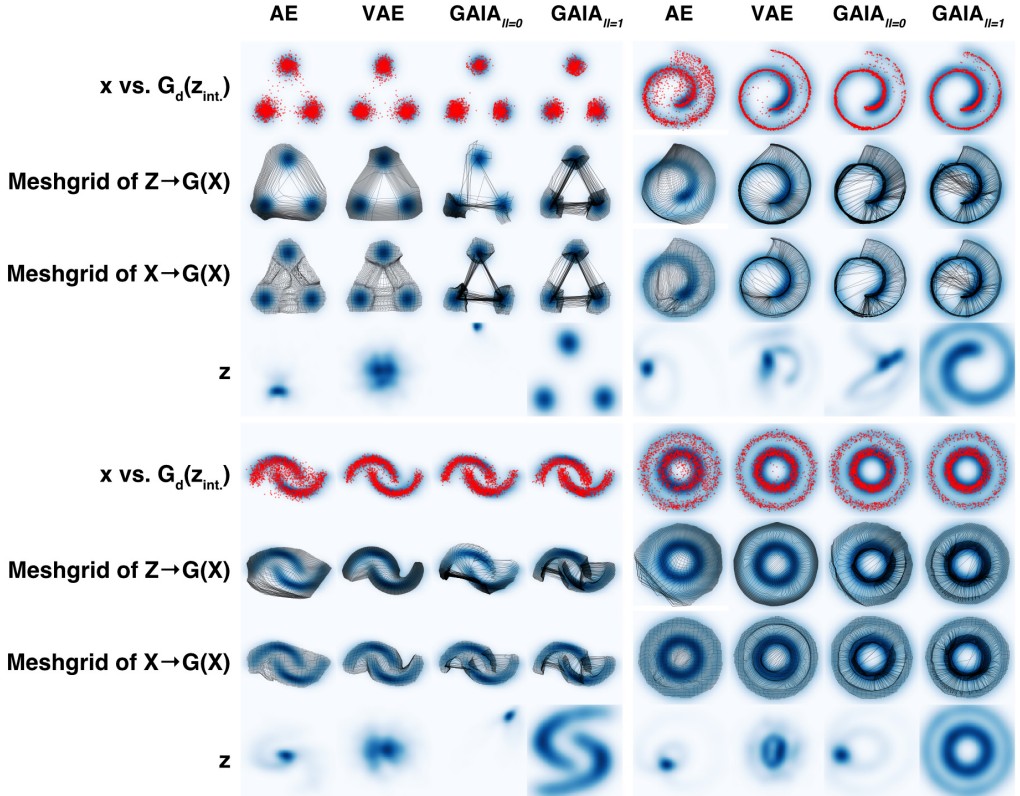

Figure 6: The same plots as in Figure 2, with the remaining four datasets.

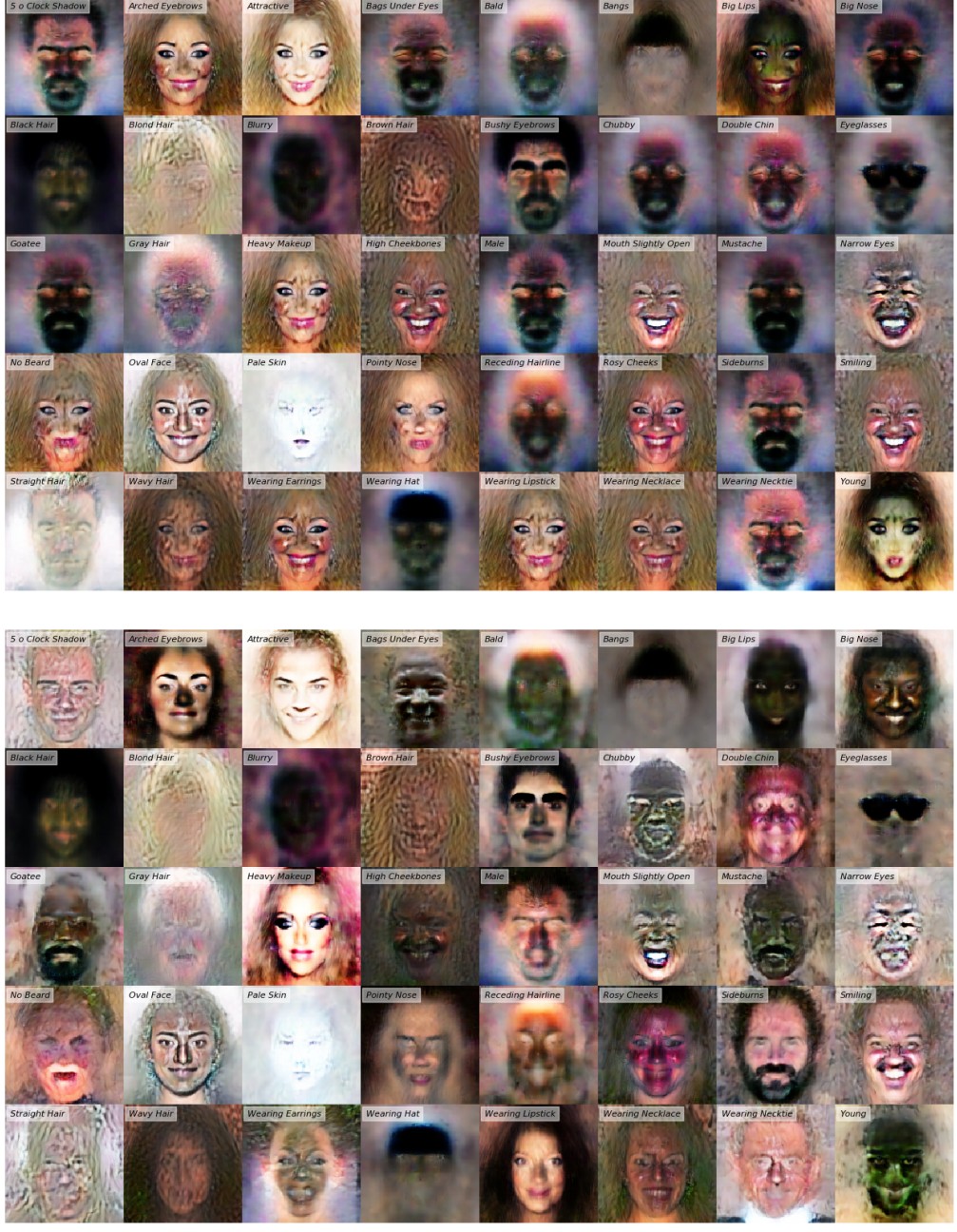

Figure 7: Attribute vectors passed through the decode of the network. Attributes on the top are found by subtracting means. Attributes on the bottom are found using ordinary least-squares regression. The attribute vectors on the bottom are more variable (for example, see Male vs Goatee vs Moustache in both panels). Zoom-in to see labels.

---

**Algorithm 1** Training the GAIA Model

---

1: $\theta_{Gen}, \theta_{Disc} \leftarrow$ initialize network parameters
2: **repeat**
3:      $x \leftarrow$ random mini-batch from dataset
4:      # pass through network
5:      $z \leftarrow G_e(x)$
6:      $z_{int.} \leftarrow interpolate(z)$
7:      $x_{gen} \leftarrow G_d(z)$
8:      $x_{int.} \leftarrow G_d(z_{int.})$
9:      $\tilde{x} \leftarrow D(x)$
10:      $\tilde{x}_{int.} \leftarrow D(x_{int.})$
11:      $\tilde{x}_{gen} \leftarrow D(x_{gen})$
12:      # compute losses
13:      $L_x \leftarrow \|x - \tilde{x}\|_1$
14:      $L_{x_{gen}} \leftarrow \|x - \tilde{x}_{gen}\|_1$
15:      $L_{x_{int.}} \leftarrow \|x_{int.} - \tilde{x}_{int.}\|_1$
16:      $L_{distance} \leftarrow pairwise distance(x, z_{gen})$
17:      # balance losses
18:      $\delta_{Disc} \leftarrow sigmoid(L_x - mean(L_{x_{gen}}, L_{x_{int.}}))$
19:      $\delta_{Gen} \leftarrow 1 - \delta_{Disc}$
20:      $\mathcal{W}_{Gen_{int.}} \leftarrow sigmoid(L_{x_{int.}} - L_{x_{gen}})$
21:      $\mathcal{W}_{Gen_{gen}} \leftarrow 1 - \mathcal{W}_{Gen_{int.}}$
22:      $\mathcal{W}_{Disc_{fake}} \leftarrow sigmoid(mean(L_{x_{gen}}, L_{x_{int.}}) \cdot \gamma - L_x)$
23:      # update parameters according to gradients
24:      $\theta_{Gen} \overset{+}{\leftarrow} -\Delta_{\theta_{Gen}}(L_{x_{gen}} \cdot \mathcal{W}_{Gen_{gen}} + L_{x_{int.}} \cdot \mathcal{W}_{Gen_{int.}} + L_{distance} * \alpha) \cdot \delta_{Disc}$
25:      $\theta_{Disc} \overset{+}{\leftarrow} -\Delta_{\theta_{Disc}}(L_x - mean(L_{x_{gen}}, L_{x_{int.}}) \cdot \mathcal{W}_{Disc_{fake}}) \cdot \delta_{Gen}$
26: **until** deadline

---

