# OpenReview forum: "Generative adversarial interpolative autoencoding: adversarial training on latent space interpolations encourages convex latent distributions"
_ICLR.cc/2019/Conference_

### Official Review · AnonReviewer1 · 2018-10-27

**Rating:** 4
**Confidence:** 5

**Review:**

This paper proposes an autoencoder architecture and training procedure for producing high-quality reconstructions and realistic interpolations. A "generator" autoencoder is trained to fool a "discriminator" autoencoder. The generator tries to minimize its own reconstruction error and minimize the reconstruction error of the discriminator when fed with interpolated latent vectors of real datapoints. The discriminator autoencoder has three losses, corresponding to minimizing reconstruction error on real datapoints and maximizing reconstruction error on the generator's output on both real datapoints and interpolated outputs. The authors also propose a loss which encourages the distances between real datapoints and their corresponding latent vectors to be similar, as well as a heuristic procedure for stabilizing GAN training. Qualitative results are shown on CelebA.

While the results look nice, the paper is not fit for publication in its current form. At a high level, the issues include a lack of convincing experimental verification of the method, a generally contradictory and confusing description of the methods, and frequent factual errors or mischaracterizations. Here I will try to describe many of the issues I found while reading the paper:
- Experimental results are only given on CelebA which is a dataset with a very strong and easy-to-model structure. The experimental results are completely qualitative. No effort is made to provide a quantitative proof of claims such as "the reconstructions are less blurry" or "the interpolations are higher quality"; only a few examples are shown. The experiments are not even described in the text, and many of the figures are unreferenced. No ablation studies are done to determine the importance of different loss terms, such as L_dist. No mention is given to how hyperparameters like alpha should be chosen (and in fact, the value given for it "1^{-4}/2" is nonsense; 1^{-4} is just 1). No results for a baseline autoencoder (i.e., just optimizing reconstruction loss) are given.
- At a higher level, no effort is given to argue why interpolation is a useful characteristic to try to encourage. There are no downstream applications proposed or tested. Earlier models, such as VAEGAN, also give reasonable reconstructions and good interpolations. Why is GAIA better? On what problem would I use GAIA and achieve better results apart from making nice-looking interpolations of people's faces?
- Definitions are often unclear or contradictory. For example, the generator autoencoder is alternatingly treating as taking input X and taking input Z. I believe what is meant is that the generator consists of two networks which compute Z = encoder(X) and X = decoder(Z). Instead, the paper just switches between G(Z) and G(X) wherever convenient. Similarly, the equation for \delta_Disc is different in Algorithm 1 and in the equation in 2.2. Interpolation, arguably one of the core parts of the model, is described as "interpolations are Euclidean interpolations between pairs of points in Z, sampled from a Gaussian distribution around the midpoint between Zg1en and Zg2en." I assume the mean of this Gaussian is the midpoint; what is its covariance? Etc.
- All autoencoders are not generative models, and in particular GAIA is not a generative model. There is no generative process. It does not estimate a data distribution. A VAE is a generative model which an autoencoder-like structure, but this does not make all autoencoders generative models.
- GAIA is described as encouraging "convex latent distributions" and a convex set is defined in the text as "A convex set of points is defined as a set in which the line connecting any pair of points will fall within the rest of the set." A convex set is not defined in terms of lines; it's defined in terms of convex combinations of points within the set. In the paper, only lines between points are considered. Claiming that the latent space is "convex" in the sense of purple blobs in B is not done - you would need to take a convex combination of multiple latent vectors and decode the results.

This is an incomplete list of the issues with this paper. The paper would need significant changes before publication.

---

> ### Author Response · Authors · 2018-11-15
> **Thank you for your comprehensive review (1/3)**
>
> Dear reviewer,
>
> We thank you for your comprehensive list of issues raised with the initial submission of our article. We believe we have exhaustively addressed each issue raised by each reviewer in our revised submission. In response to each reviewer, we have divided each review into a point-by-point list of each issue raised followed by our response, pointing to where that issue was addressed in the text.
>
> We have made several major revisions, listed below, as well as a number of other revisions which are addressed point-by-point in response to reviewers.
>
> Major revisions:
> As requested by all three reviewers, we added a set of quantitative and ablation experiments on a low dimensional dataset. These experiments can be seen in Figures 2 and 6, as well as Table 1.
> We added an experiments section to the text and rearranged the text for structure.
> We rewrote sections of the introduction to better motivate our research.
> We added a number of relevant references and extended our discussion of related works.
> We edited the entire document for consistent notation both internally and to other related papers.
>
> We thank you for the time and energy put into your excellent reviews of our article and believe that our submission has greatly increased in quality because of your input.
>
> ________________________________________
> "- Experimental results are only given on CelebA which is a dataset with a very strong and easy-to-model structure. The experimental results are completely qualitative.
> -------------------------------------------------------
> We added a low dimensional dataset example (Figure 2, Table 1, Figure 6), and a quantitative assessment of the likelihood of interpolations and reconstructions, the correlation between latent structure and structure in high dimensional space, and the KL divergence between data and interpolations, as well as data and reconstructions. We compared a VAE, and AE, and GAIA (both with and without the pairwise-distance loss term).
>
> ________________________________________
> "No effort is made to provide a quantitative proof of claims such as "the reconstructions are less blurry" or "the interpolations are higher quality"; only a few examples are shown.
> -------------------------------------------------------
> We edited these section and sentences to not make comparative claims. We now show examples in the appendix of reconstructions on several low dimensional datasets. We can add additional interpolation and attribute vector figures in the appendix as well if the reviewer finds it important.
>
> ________________________________________
> "The experiments are not even described in the text, and many of the figures are unreferenced.
> -------------------------------------------------------
> We added an experiments section to the text which now includes subsections for both datasets, describing each measure.
>
> ________________________________________
> "No ablation studies are done to determine the importance of different loss terms, such as L_dist. No mention is given to how hyperparameters like alpha should be chosen
> -------------------------------------------------------
> We now perform an ablation study by comparing our autoencoder with adversarial regularization, to an autoencoder of the same architecture without regularization. We also ablate L_dist.

---

> > ### Author Response · Authors · 2018-11-15
> > **Thank you for your comprehensive review (2/3)**
> >
> > ________________________________________
> > " (and in fact, the value given for it "1^{-4}/2" is nonsense; 1^{-4} is just 1). No results for a baseline autoencoder (i.e., just optimizing reconstruction loss) are given.
> > -------------------------------------------------------
> > Thank you for pointing out this typo. We added results for a baseline autoencoder in the low dimensional datasets in Figures 2 and 6 as well as Table 1.
> >
> > ________________________________________
> > "- At a higher level, no effort is given to argue why interpolation is a useful characteristic to try to encourage. There are no downstream applications proposed or tested. Earlier models, such as VAEGAN, also give reasonable reconstructions and good interpolations. Why is GAIA better? On what problem would I use GAIA and achieve better results apart from making nice-looking interpolations of people's faces?
> > -------------------------------------------------------
> > We discuss in the introduction that interpolations train subsets of the convex hull of the distribution to produce realistic samples in X. We expanded upon a discussion of downstream applications in psychophysics and neuroscience, which we believe is sufficient motivation for our work. We reference and discuss VAE-GAN in Figure 5, as well as the related works section. We mention that our network is based upon aspects VAE-GAN but takes a different approach at reconstruction at a pixel-level rather than at an abstract level (VAEGAN uses a reconstruction loss in a latent layer of the discriminator rather than pixel-level loss).
> >
> > ________________________________________
> > "- Definitions are often unclear or contradictory. For example, the generator autoencoder is alternatingly treating as taking input X and taking input Z. I believe what is meant is that the generator consists of two networks which compute Z = encoder(X) and X = decoder(Z). Instead, the paper just switches between G(Z) and G(X) wherever convenient. Similarly, the equation for \delta_Disc is different in Algorithm 1 and in the equation in 2.2.
> > -------------------------------------------------------
> > We edited the entire document for notation both more internally consistent, and consistent with other related works. We now mention in a footnote in section 1 that G(x_i) is equivalent to (G_d(G_e(x_i)), and now write G(Z) as G_d(Z).
> >
> > ________________________________________
> > " Interpolation, arguably one of the core parts of the model, is described as "interpolations are Euclidean interpolations between pairs of points in Z, sampled from a Gaussian distribution around the midpoint between Zg1en and Zg2en." I assume the mean of this Gaussian is the midpoint; what is its covariance? Etc.
> > -------------------------------------------------------
> > We expanded upon our explanation of the univariate Gaussian used to sample midpoints, including its standard deviation, in section 2.
> >
> > ________________________________________
> > "- All autoencoders are not generative models, and in particular GAIA is not a generative model. There is no generative process. It does not estimate a data distribution. A VAE is a generative model which an autoencoder-like structure, but this does not make all autoencoders generative models.
> > -------------------------------------------------------
> > We added a discussion in section 1.1 (paragraph 2) about generative models. We make it more explicit in this section that GAIA is not a generative model.
> >
> > ________________________________________

---

> > > ### Author Response · Authors · 2018-11-15
> > > **Thank you for your comprehensive review (3/3)**
> > >
> > > "- GAIA is described as encouraging "convex latent distributions" and a convex set is defined in the text as "A convex set of points is defined as a set in which the line connecting any pair of points will fall within the rest of the set." A convex set is not defined in terms of lines; it's defined in terms of convex combinations of points within the set. In the paper, only lines between points are considered. Claiming that the latent space is "convex" in the sense of purple blobs in B is not done - you would need to take a convex combination of multiple latent vectors and decode the results.
> > > -------------------------------------------------------
> > > We added a citation (Klee, 1974, “What is a convex set?”) that uses similar wording to ours to describe a convex set.
> > >
> > > “ Though convex sets are defined in various settings (see [27] for a survey), the most useful definitions are based on a notion of betweenness. When E is a space in which such a notion is defined, a subset C of E is called convex provided that for each two points x and y of C, C includes all points between x and y. The most important setting, and the only one to be discussed here, is that in which E is a vector space over the real number field R or, in particular, is the n-dimensional Euclidean space En, and the points between x and y are those of the line segment xy. Thus, a subset C of a real vector space is convex provided that C contains every segment whose endpoints both belong to C. “
> > >
> > > We also recognize the reviewer's claims as being valid. We have reworded the introduction to say that we use interpolations in latent space to explicitly train on at least some subsets of the convex hull of the latent distribution. Figure 1D is also intended to show that not all regions of the convex hull of the dataset in latent space will be reached with our method. We also added a discussion to the conclusions section calling for future work to train upon the full convex hull of the latent distribution.
> > >
> > > ________________________________________
> > > "This is an incomplete list of the issues with this paper. The paper would need significant changes before publication.
> > > -------------------------------------------------------
> > > We hope the reviewer agrees that we have made significant changes to this paper, in the form of additional experiments, quantitative results, clarifications, and addressing point-by-point the concerns and issues raised by each reviewer. We thank the reviewer for their reconsideration of our paper in light of these changes - and ask them to update us with a complete list of their remaining issues.

---

### Official Review · AnonReviewer2 · 2018-11-01

**Rating:** 4
**Confidence:** 4

**Review:**


Update:

I’d like to thank the authors for their thoroughness in responding to the issues I raised. I will echo my fellow reviewers in saying that I would encourage the authors to submit to another venue, given the substantial modifications made to the original submission.

The updated version provides a clearer context for the proposed approach (phychophysical experimentation) and avoids mischaracterizing GAIA as a generative model.

Despite more emphasis being put on mentioning the existence of bidirectional variants of GANs, I still feel that the paper does not adequately address the following question: “What does GAIA offer that is not already achievable by models such as ALI, BiGAN, ALICE, and IAE, which equip GANs with an inference mechanism and can be used to perform interpolations between data points and produce sharp interpolates?” To be clear, I do think that the above models are inadequate for the paper’s intended use (because their reconstructions tend to be semantically similar but noticeably different perceptually), but I believe this is a question that is likely to be raised by many readers.

To answer the authors’ questions:

- Flow-based generative models such as RealNVP relate to gaussian latent spaces in that they learn to map from the data distribution to a simple base distribution (usually a Gaussian distribution) in a way that is invertible (and which makes the computation of the Jacobian’s determinant tractable). The base distribution can be seen as a Gaussian latent space which has the same dimensionality as the data space.
- Papers on building more flexible approximate posteriors in VAEs: in addition to the inverse autoregressive flow paper already cited in the submission, I would point the authors to Rezende and Mohamed’s “Variational Inference with Normalizing Flows”, Huang et al.’s “Neural Autoregressive Flows”, and van den Berg et al.’s “Sylvester Normalizing Flows for Variational Inference”.

-----

The paper title summarizes the main claim of the paper: "adversarial training on latent space interpolations encourage[s] convex latent distributions". A convex latent space is defined as a space in which a linear interpolation between latent codes obtained by encoding a pair of points from some data distribution yields latent codes whose decoding also belongs to the same data distribution. The authors argue that current leading approaches fall short of producing convex latent spaces while preserving the "high-dimensional structure of the original distribution". They propose a GAN-AE hybrid, called GAIA, which they claim addresses this issue. The proposed approach turns the GAN generator and discriminator into autoencoders, and the adversarial game is framed in terms of minimizing/maximizing the discriminator’s reconstruction error. In addition to that, interpolations between pairs of data points are computed in the generator’s latent space, and the interpolations are decoded and treated as generator samples. A regularization term is introduced to encourage distances between pairs of data points to be mirrored by their representation in the generator’s latent space. The proposed approach is evaluated through qualitative inspection of latent space interpolations, attribute manipulations, attribute vectors, and generator reconstructions.

Overall I feel like the problem presented in the paper is well-justified, but the paper itself does not build a sufficiently strong argument in favor of the proposed approach for me to recommend its acceptance. I do think there is a case to be made for a model which exhibits sharp reconstructions and which allows realistic latent space manipulations -- and this is in some ways put forward in the introduction -- but I don’t feel that the way in which the paper is currently cast highlights this very well. Here is a detailed breakdown of why, and where I think it should be improved, roughly ordered by importance:

- The main reason for my reluctance to accept the paper is the fact that its main subject is convex latent spaces, yet I don’t see that reflected in the evaluation. The authors do not address how to evaluate (quantitatively or qualitatively) whether a certain model exhibits a convex latent space, and how to compare competing approaches with respect to latent space convexity. Figure 2 does present latent space interpolations which help get a sense of the extent to which interpolates also belong to the data distribution, however in the absence of a comparison to competing approaches it’s impossible for me to tell whether the proposed approach yields more convex latent spaces.
- I don’t agree with the premise that current approaches are insufficient. The authors claim that autoencoders produce blurry reconstructions; while this may be true for factorized decoders, autoregressive decoders should alleviate this issue. They also claim that GANs lack bidirectionality but fail to mention the existing line of work in that direction (ALI, BiGAN, ALICE, and more recently Implicit Autoencoders). Finally, although flow-based generative models are mentioned later in the paper, they are not discussed in Section 1.2 when potential approaches to building convex latent spaces are enumerated and declared insufficient. As a result, the paper feels a little disconnected from the current state of the generative modeling literature.
- The necessity for latent spaces to "respect the high-dimensional structure of the [data] distribution" is stated as a fact but not well-justified. How do we determine whether a marginal posterior is "a suboptimal representation of the high-dimensional dataset"? I think a more nuanced statement would be necessary. For instance, many recent approaches have been proposed to build more flexible approximate posteriors in VAEs; would that go some way towards embedding the data distribution in a more natural way?
- I also question whether latent space convexity is a property that should always hold. In the case of face images a reasonable argument can be made, but in a dataset such as CIFAR10 how should we linearly interpolate between a horse and a car?
- The proposed model is presented in the abstract as an "AE which produces non-blurry samples", but it’s not clear to me how one would sample from such a model. The generator is defined as a mapping from data points to their reconstruction; does this mean that the sampling procedure requires access to training examples? Alternatively one could fit a prior distribution on top of the latent codes and their interpolations, but as far as I can tell this is not discussed in the paper. I would like to see a more thorough discussion on the subject.
- When comparing reconstructions with competing approaches there are several confounding factors, like the resolution at which the models were trained and the fact that they all reconstruct different inputs. Removing those confounding factors by comparing models trained at the same resolution and reconstructing the same inputs would help a great deal in comparing each approach.
- The structure-preserving regularization term compares distances in X and Z space, but I doubt that pixelwise Euclidian distances are good at capturing an intuitive notion of distance: for example, if we translate an image by a few pixels the result is perceptually very similar but its Euclidian distance to the original image is likely to be high. As far as I can tell, the paper does not present evidence backing up the claim that the regularization term does indeed preserve local structure.
- Figures 2 and 3 are never referenced in the main text, and I am left to draw my own conclusions as to what claim they are supporting. As far as I can tell they showcase the general capabilities of the proposed approach, but I would have liked to see a discussion of whether and how they improve on results that can be achieved by competing approaches.
- The decision of making the discriminator an autoencoder is briefly justified when discussing related work; I would have liked to see a more upfront and explicit justification when first introducing the model architecture.
- When discussing feature vectors it would be appropriate to also mention Tom White’s paper on Sampling Generative Networks.

---

> ### Author Response · Authors · 2018-11-15
> **Thank you for your comprehensive review (1/3)**
>
> Dear reviewer,
>
> We thank you for your comprehensive list of issues raised with the initial submission of our article. We believe we have exhaustively addressed each issue raised by each reviewer in our revised submission. In response to each reviewer, we have divided each review into a point-by-point list of each issue raised followed by our response, pointing to where that issue was addressed in the text.
>
> We have made several major revisions, listed below, as well as a number of other revisions which are addressed point-by-point in response to reviewers.
>
> Major revisions:
> As requested by all three reviewers, we added a set of quantitative and ablation experiments on a low dimensional dataset. These experiments can be seen in Figures 2 and 6, as well as Table 1.
> We added an experiments section to the text and rearranged the text for structure.
> We rewrote sections of the introduction to better motivate our research.
> We added a number of relevant references and extended our discussion of related works.
> We edited the entire document for consistent notation both internally and to other related papers.
>
> We thank you for the time and energy put into your excellent reviews of our article and believe that our submission has greatly increased in quality because of your input.
>
> ________________________________________
> "- The main reason for my reluctance to accept the paper is the fact that its main subject is convex latent spaces, yet I don’t see that reflected in the evaluation. The authors do not address how to evaluate (quantitatively or qualitatively) whether a certain model exhibits a convex latent space, and how to compare competing approaches with respect to latent space convexity. Figure 2 does present latent space interpolations which help get a sense of the extent to which interpolates also belong to the data distribution, however in the absence of a comparison to competing approaches it’s impossible for me to tell whether the proposed approach yields more convex latent spaces.
> -------------------------------------------------------
> We added an experiment section in which we trained GAIA, and AE, and a VAE on 2D several distributions (Section 3.1). In this experiment, we show qualitative and quantitative results which confirm that GAIA is learning to yield interpolated samples which more directly model the true data distribution in X.
>
> ________________________________________
> "- I don’t agree with the premise that current approaches are insufficient. The authors claim that autoencoders produce blurry reconstructions; while this may be true for factorized decoders, autoregressive decoders should alleviate this issue.
> -------------------------------------------------------
> We have specified that we are referring to certain subsets of autoencoders. We expanded upon our discussions in related works as well as the introduction about alternative approaches.

---

> > ### Author Response · Authors · 2018-11-15
> > **Thank you for your comprehensive review (2/3)**
> >
> > ________________________________________
> > " They also claim that GANs lack bidirectionality but fail to mention the existing line of work in that direction (ALI, BiGAN, ALICE, and more recently Implicit Autoencoders).
> > -------------------------------------------------------
> > We would like to thank the reviewer for pointing out Implicit Autoencoders, which we have now included in our discussions of related work and included in our comparison of Figure 4. We would also like to note that the other papers mentioned are cited in Figure 5, and referenced in section 2.5. We expanded upon our related works section as well, as noted above.
> >
> > ________________________________________
> > " Finally, although flow-based generative models are mentioned later in the paper, they are not discussed in Section 1.2 when potential approaches to building convex latent spaces are enumerated and declared insufficient. As a result, the paper feels a little disconnected from the current state of the generative modeling literature.
> > -------------------------------------------------------
> > We expanded upon our discussion in related works and the introduction and added references to autoregressive models as a second alternative to GANs. We did not add a discussion of flow-based generative models to Section 1.2 as we were unsure of how flow-based models relate to latent space convexity. We would also be open to discussing flow-based models in 1.2 if the reviewer could elaborate on why flow-based models belong there.
> >
> > ________________________________________
> > "- The necessity for latent spaces to "respect the high-dimensional structure of the [data]  distribution" is stated as a fact but not well-justified. How do we determine whether a marginal posterior is "a suboptimal representation of the high-dimensional dataset"? I think a more nuanced statement would be necessary. For instance, many recent approaches have been proposed to build more flexible approximate posteriors in VAEs; would that go some way towards embedding the data distribution in a more natural way?
> > -------------------------------------------------------
> > We modified this statement to mention that an ongoing effort in generative modeling has been to build more flexible posteriors in VAEs. We invite the reviewer to direct us to any examples of building more flexible posteriors. We agree with the reviewer that work in this direction could be important for embedding the data distribution in a more natural way while still being generative.
> >
> > ________________________________________
> > "- I also question whether latent space convexity is a property that should always hold. In the case of face images a reasonable argument can be made, but in a dataset such as CIFAR10 how should we linearly interpolate between a horse and a car?
> > -------------------------------------------------------
> > We agree that with the reviewer that latent space convexity is not necessarily a desirable property in all applications, which we discuss in on-going work. However, in some applications (as discussed in the work cited in psychophysics), we believe that a convex latent space has some utility (e.g. in producing samples which are perceptually realistic).
> >
> > ________________________________________
> > "- The proposed model is presented in the abstract as an "AE which produces non-blurry samples", but it’s not clear to me how one would sample from such a model. The generator is defined as a mapping from data points to their reconstruction; does this mean that the sampling procedure requires access to training examples? Alternatively one could fit a prior distribution on top of the latent codes and their interpolations, but as far as I can tell this is not discussed in the paper. I would like to see a more thorough discussion on the subject.
> > -------------------------------------------------------
> > The reviewer is correct that while a prior distribution could be fit on top of the latent codes and their interpolations, the proposed model does not include a prior to sample from. As such, the model is not by definition a generative model. We added an explanation of this in the introduction.

---

> > > ### Author Response · Authors · 2018-11-15
> > > **Thank you for your comprehensive review (3/3)**
> > >
> > > ________________________________________
> > > "- When comparing reconstructions with competing approaches there are several confounding factors, like the resolution at which the models were trained and the fact that they all reconstruct different inputs. Removing those confounding factors by comparing models trained at the same resolution and reconstructing the same inputs would help a great deal in comparing each approach.
> > > -------------------------------------------------------
> > > We are choosing not to run each network individually at the same resolution because of the computational resources and time that would be required to set up each model. Instead, we have moved the comparison to the appendix and increased and discuss in the figure caption differences in network architecture, image dataset, and image resolution. We also note that these images should not be compared for quality, but for the similarity between input and reconstruction.
> > >  If the reviewer has an idea for how we could perform a better comparison we would be open to suggestions.
> > >
> > > ________________________________________
> > > "- The structure-preserving regularization term compares distances in X and Z space, but I doubt that pixelwise Euclidian distances are good at capturing an intuitive notion of distance: for example, if we translate an image by a few pixels the result is perceptually very similar but its Euclidian distance to the original image is likely to be high. As far as I can tell, the paper does not present evidence backing up the claim that the regularization term does indeed preserve local structure.
> > > -------------------------------------------------------
> > > We agree that a pixel-wise Euclidean distance preservation term is not always necessarily a good measure of distance. In our new experiments, we compare the network with and without this regularization. We now have a metric (correlation) for the preservation of local structure compared in Table 1.
> > >
> > > ________________________________________
> > > "- Figures 2 and 3 are never referenced in the main text, and I am left to draw my own conclusions as to what claim they are supporting. As far as I can tell they showcase the general capabilities of the proposed approach, but I would have liked to see a discussion of whether and how they improve on results that can be achieved by competing approaches.
> > >
> > > We now reference each figure in the text. We now use Table 1 as a comparison against AE and VAE, which to us were the most relevant baseline models to compare against.
> > >
> > > ________________________________________
> > > "- The decision of making the discriminator an autoencoder is briefly justified when discussing related work; I would have liked to see a more upfront and explicit justification when first introducing the model architecture.
> > > -------------------------------------------------------
> > > We discuss the decision to make the discriminator an autoencoder is now discussed in the third paragraph in section 2.0, the last paragraph of section 2.0, and the first paragraph of the conclusion.
> > >
> > > ________________________________________
> > > "- When discussing feature vectors it would be appropriate to also mention Tom White’s paper on Sampling Generative Networks.
> > > -------------------------------------------------------
> > > We thank the reviewer for this suggestion. We have integrated the feature vector method from this paper into our discussion in several sections of the paper, including the feature vector section.

---

> > > > ### Comment · AnonReviewer1 · 2018-11-15
> > > > **Feedback on update**
> > > >
> > > > First, I appreciate the care you took in updating your paper and the comprehensiveness of your response. I think the draft is clearly improved, but I want to be clear up front that I think the changes required to make this paper suitable for acceptance are beyond the threshold specified by the call for papers, which states that "Area chairs and reviewers reserve the right to ignore changes that are significantly different from the original paper."
> > > >
> > > > I have reviewed the updated draft. I appreciate the addition of the synthetic datasets in order to provide a more quantitative analysis, the inclusion of additional baselines, and the efforts to clarify notation. However, there remain many significant issues with the paper.
> > > > - The description of how the interpolation coefficient \beta is sampled (to produce z_int) is still incomplete and doesn't make sense -- it is written as \beta \sim N(\mu, \sigma^2). \mu is not defined. Note that the support of the Gaussian is the entire real number line, suggesting that \beta can be negative, with the extent depending on \mu. Letting \beta be negative does not make sense to me.
> > > > - The paper states that "five" 2D datasets are used. What are they? I only see mention of the "S" dataset explicitly. How were the results on the different datasets combined to produce Table 1?
> > > > - I assume that when you are computing r(x, z) you are seeing whether the structure of the latent space matches the structure of the real data. I can see why this might be desirable on a simple 2D dataset like the "S" dataset, but in general this may be the *opposite* of what we want out of an autoencoder! To produce a useful representation (and to produce nice interpolations), the latent space can often need to be dramatically "warped" compared to the data space.
> > > > - You never define what exactly a "meshgrid" is or how the meshgrid of z -> g(x) is different from x -> g(x) for GAIA.
> > > > - As a final note, I maintain that CelebA is not a difficult dataset because the images are carefully normalized (so the "underlying data manifold" is not complex compared to, say, CIFAR-10 or ImageNet). Further, the results on CelebA are impressive but it's not at all obvious how much GAIA improves upon VAEGAN and there is simply no way (or attempt) to make this comparison more rigorous.
> > > >
> > > >  I would suggest the authors consider resubmitting the updated version (with additional changes to address my continued concerns) to a different conference.

---

### Official Review · AnonReviewer3 · 2018-11-02

**Rating:** 5
**Confidence:** 4

**Review:**

Update:

I appreciate the effort put by the authors into improving the paper. The revised draft is much better than the initial one. But I agree to AnonReviewer2 in that the degree to which this paper has to be modified goes beyond what the review process (even at ICLR) assumes. It is wrong to submit a very unfinished paper and then use the review period to polish it and add results. This incurs unnecessary extra load on the review process.

The added 2D results are toy-ish and somewhat confusing (I am not sure I understand what the meshgrids are and what do they tell us). Generally, some toy examples are good to illustrate the method, but they are not enough as a serious evaluation. The paper should have more results on complex datasets, like for instance ImageNet or LSUN or CIFAR or so, and should have comparisons to existing VAE-GAN hybrids, like VAE-GAN. Also, since a lot of the authors’ motivation seems to come from psychophysics, showing some application to that might be a good way to showcase the value of the method (although this may not go well if submitting to machine learning conferences).

I encourage the authors to further strengthen the paper and resubmit to another venue.

-----

The paper proposes a model for image generation that combines an autoencoder with a generative adversarial network (GAN). The GAN is used to enforce that interpolations between latent vectors of two samples from the training set are decoded to realistic images. The method is applied to attribute manipulation and interpolation of face images.

Pros:
1) A simple and reasonable formulation
2) Visually good reconstruction of samples and convincing interpolation between samples on the CelebA-HQ dataset.
3) Good qualitative facial attribute manipulation results on the CelebA-HQ dataset.

Cons:
1) Experimental evaluation is very limited. There is just one dataset and only qualitative results. This is unacceptable: the method should be evaluated on more datasets and there should be quantitative results. I do realize it is not trivial to get quantitative, but it is possible. For instance, a user study can always be performed. But I believe one could also come up with simpler-to-measure metrics for at least some of the reported tasks. There is no comparison to other methods for facial attribute manipulation (for instance, StarGAN).
2) There is no ablation study. To which extent is each of the model components important? For instance, the interpolation adversarial loss, the discriminator/generator balancing term, the network architecture (autoencoder discriminator)?
3) As I understand, it is impossible to randomly sample directly from the model, only interpolate/modify existing images. This is a difference from most of prior work. It should be discussed clearly.
4) Related work discussion is quite brief and misses some relevant work, for instance Adversarial Autoencoders (Makhazani et al., ICLR 2016, somewhat related) or Adversarially Constrained Autoencoder Interpolations (Berthelot et al., arxiv 2018, it’s concurrent, but could be good to discuss).
5) Writing is not of very high quality. There are typos, grammatical issues, and questionable statements. The manuscript should be significantly improved. Specific comments:
- The structure is quite strange. There is no separation between the method and the experiments, the related work comes in very late and is very brief. This is all not critical, but confusing.
- A typo in the title: should be “encourages”
- Second sentence of the introduction should be supported with evidence (for instance references)
- “Two unsupervised neural network based algorithms, the Autoencoder (AE; Hinton & Salakhutdinov
2006) and Generative Adversarial Network (GAN; Goodfellow et al. 2014), are at present the most popular tools in generative modeling.” - Vanilla autoencoders are not very popular tools for generative modeling. Variational autoencoders and some other flavors are.
- “Unsupervised neural network approaches are often ideal generative models because they require little or no tweaking of network architectures to handle entirely new datasets.” I do not really get this sentence. What is the alternative to unsupervised generative models? Why do unsupervised approaches not require tweaking? (In my experience, they very well benefit from tweaking.)
- “… a lack of certain constraints on the generative capacity of current neural-network based generative models make it challenging to infer structure from their latent generative representations.” What does “a lack of certain constraints” mean? There are some constraints, for instance the latent space is usually forced to correspond to a fixed distribution. Moreover, there is a lot of work on disentangling that also aims to find structure in latent spaces (for instance, InfoGAN).
- “and promotes convexity in the model’s generative capacity.” What is convexity in the capacity? I do not think this is grammatical.
- In 1.1 the mathematical notation seems wrong. Does X really denote a set? In what follows it seems that X is used interchangeably for three different things: a sample from the dataset, the set of training samples, and the space the samples come from.
- “bidirectionality of adversarially generated data.” What is bidirectional data?
- “AEs tend to produce blurry images due to their pixel-wise error functions (Larsen et al., 2015)” Perhaps this was intended to refer to VAEs. AEs can generate perfectly sharp images if given enough capacity.
- “method more greatly resembles the original data than other GAN-based methods” Method does not resemble data
- “Due to their exact latent-variable inference, these architectures may also provide a useful direction for developing generative models to explore latent-spaces of data for generating datasets for psychophysical experiments.” This is mentioned a few times, but never supported
- Acknowledgements should not be in the review version (can violate anonymity)

5) Minor: Why is a Gaussian around the midpoint used for interpolations? Why not all convex combinations of two, or possibly more, samples?

To conclude, the paper presents quite good qualitative results on the CelebA-HQ dataset, but has problems with the thoroughness of the experimental evaluation, discussion of the related work, and presentation. The paper cannot be published in its current form.

---

> ### Author Response · Authors · 2018-11-15
> **Thank you for your comprehensive review (1/3)**
>
> Dear reviewer,
>
> We thank you for your comprehensive list of issues raised with the initial submission of our article. We believe we have exhaustively addressed each issue raised by each reviewer in our revised submission. In response to each reviewer, we have divided each review into a point-by-point list of each issue raised followed by our response, pointing to where that issue was addressed in the text.
>
> We have made several major revisions, listed below, as well as a number of other revisions which are addressed point-by-point in response to reviewers.
>
> Major revisions:
> As requested by all three reviewers, we added a set of quantitative and ablation experiments on a low dimensional dataset. These experiments can be seen in Figures 2 and 6, as well as Table 1.
> We added an experiments section to the text and rearranged the text for structure.
> We rewrote sections of the introduction to better motivate our research.
> We added a number of relevant references and extended our discussion of related works.
> We edited the entire document for consistent notation both internally and to other related papers.
>
> We thank you for the time and energy put into your excellent reviews of our article and believe that our submission has greatly increased in quality because of your input.
>
> ________________________________________
> "1) Experimental evaluation is very limited. There is just one dataset and only qualitative results. This is unacceptable: the method should be evaluated on more datasets and there should be quantitative results. I do realize it is not trivial to get quantitative, but it is possible. For instance, a user study can always be performed. But I believe one could also come up with simpler-to-measure metrics for at least some of the reported tasks. "
> -------------------------------------------------
> We have performed additional experiments on lower-dimensional (2D) datasets comparing our results to vanilla autoencoders and variational autoencoders. These experiments were added to section 3.1, see Figures 2 and 6, as well as and Table 1.
> ________________________________________
> "There is no comparison to other methods for facial attribute manipulation (for instance, StarGAN)."
> ------------------------------------------------
> We added a discussion to section 3.2.1 about prior work on attribute manipulation. We discuss models which utilize feature information during learning (e.g. StarGAN, CycleGAN, etc) and models in which latent features are not part of learning.
>
> ________________________________________
> "2) There is no ablation study. To which extent is each of the model components important? For instance, the interpolation adversarial loss, the discriminator/generator balancing term, the network architecture (autoencoder discriminator)?"
> ------------------------------------------------
> We have added ablation studies of the lower-dimensional datasets. In particular, we ablate (1) the multidimensional scaling error function and find that pairwise distance in Z vs X is impacted, and we able (2) we effectively ablate the adversarial loss function from GAIA (by comparing a vanilla AE). We find the generated likelihood of generated latent interpolations in both cases decreases (Table 1).
>
> ________________________________________
> "3) As I understand, it is impossible to randomly sample directly from the model, only interpolate/modify existing images. This is a difference from most of the prior work. It should be discussed clearly."
> ------------------------------------------------
> We added a discussion in section 1.1 about the difference between generative and non-generative models. We now state explicitly that GAIA is not a generative model, and discuss the implications for random sampling.
>
> ________________________________________
> "4) Related work discussion is quite brief and misses some relevant work, for instance, Adversarial Autoencoders (Makhazani et al., ICLR 2016, somewhat related) or Adversarially Constrained Autoencoder Interpolations (Berthelot et al., arXiv 2018, it’s concurrent, but could be good to discuss)."
> ------------------------------------------------
> We significantly expanded upon our discussion of related work such as Makhazani et al (2016) in section 3.3. We also Adversarial Autoencoders and Adversarially Constrained Autoencoder Interpolations (ACAI).
>
> ________________________________________
> "5) Writing is not of very high quality. There are typos, grammatical issues, and questionable statements. The manuscript should be significantly improved. "
> ------------------------------------------------
> We have performed a significant rewrite of our paper to improve the clarity and organization of the writing.

---

> > ### Author Response · Authors · 2018-11-15
> > **Thank you for your comprehensive review (2/3)**
> >
> > ________________________________________
> > "- A typo in the title: should be “encourages”
> > -------------------------------------------------------
> > The typo in the title has been fixed.
> >
> > ________________________________________
> > "- The second sentence of the introduction should be supported with evidence (for instance references)
> > -------------------------------------------------------
> > We have added references to the second sentence. We are also considering adding a third example dataset which can be added before the final revision deadline if the reviewer finds it important for assessing the utility of our network.
> >
> > ________________________________________
> > "- “Two unsupervised neural network based algorithms, the Autoencoder (AE; Hinton & Salakhutdinov
> > 2006) and Generative Adversarial Network (GAN; Goodfellow et al. 2014), are at present the most popular tools in generative modeling.” - Vanilla autoencoders are not very popular tools for generative modeling. Variational autoencoders and some other flavors are.
> > -------------------------------------------------------
> > We have specified that certain classes of neural networks are popular generative modeling tools. We also updated this sentence and section to include autoregressive and flow based models.
> >
> > ________________________________________
> > "- “Unsupervised neural network approaches are often ideal generative models because they require little or no tweaking of network architectures to handle entirely new datasets.” I do not really get this sentence. What is the alternative to unsupervised generative models? Why do unsupervised approaches not require tweaking? (In my experience, they very well benefit from tweaking.)
> > -------------------------------------------------------
> > We removed this sentence and instead discuss the utility of unsupervised approaches to learning feature spaces over hand-engineering approaches.
> >
> > ________________________________________
> > "- “… a lack of certain constraints on the generative capacity of current neural-network based generative models make it challenging to infer structure from their latent generative representations.” What does “a lack of certain constraints” mean? There are some constraints, for instance the latent space is usually forced to correspond to a fixed distribution. Moreover, there is a lot of work on disentangling that also aims to find structure in latent spaces (for instance, InfoGAN).
> > -------------------------------------------------------
> > We removed this sentence from section 1.1 and rewrote parts of the final paragraph.
> >
> > ________________________________________
> > "- “and promotes convexity in the model’s generative capacity.” What is convexity in the capacity? I do not think this is grammatical.
> > -------------------------------------------------------
> > We reworded the final paragraph of section 1, to state: “We propose a novel AE that hybridizes features of an AE and a GAN. Our network is trained explicitly to control for the structure of latent representations, and promotes convexity in latent space by using adversarial constraints on interpolations between data samples in latent space to produce realistic samples within at least some subsets of the convex hull of the latent distribution.“
> >
> > ________________________________________
> > "- In 1.1 the mathematical notation seems wrong. Does X really denote a set? In what follows it seems that X is used interchangeably for three different things: a sample from the dataset, the set of training samples, and the space the samples come from.
> > -------------------------------------------------------
> > We edited the entire document for notation both more internally consistent, and consistent with other related works.
> >
> > ________________________________________
> > "- “bidirectionality of adversarially generated data.” What is bidirectional data?
> > -------------------------------------------------------
> > We removed this sentence and rewrote sections of our manuscript to be more explicit with our discussion of bidirectionality.
> >
> > ________________________________________
> > "- “AEs tend to produce blurry images due to their pixel-wise error functions (Larsen et al., 2015)” Perhaps this was intended to refer to VAEs. AEs can generate perfectly sharp images if given enough capacity.
> > -------------------------------------------------------
> > We specify that we are referring to AEs which perform dimensionality reduction and certain forms of autoencoders such as VAEs in Section 1.3.

---

> > > ### Author Response · Authors · 2018-11-15
> > > **Thank you for your comprehensive review (3/3)**
> > >
> > >
> > > ________________________________________
> > > "- “method more greatly resembles the original data than other GAN-based methods” Method does not resemble data
> > > -------------------------------------------------------
> > > We reworded this section to say that our loss function allows us to resemble input data at a pixel-level. We also added to the discussion of the comparison figure (Now Figure 5 in the appendix) to discuss the differences in architectures shown.
> > >
> > > ________________________________________
> > > "- “Due to their exact latent-variable inference, these architectures may also provide a useful direction for developing generative models to explore latent-spaces of data for generating datasets for psychophysical experiments.” This is mentioned a few times, but never supported
> > > -------------------------------------------------------
> > > We added a reference here too (Kingma et al., 2018), which uses the same wording “exact latent-variable inference” as we do to describe their work. We also reference psychophysical experiments using generative networks.
> > >
> > > ________________________________________
> > > "- Acknowledgements should not be in the review version (can violate anonymity)
> > > -------------------------------------------------------
> > > We removed this section and apologize for the error.
> > >
> > > ________________________________________
> > > "5) Minor: Why is a Gaussian around the midpoint used for interpolations?
> > > -------------------------------------------------------
> > > We added a footnote discussing this in section 2. We used a Gaussian centered around the midpoint because we found that interpolations from near to the original samples required less training than interpolations near the midpoint.
> > >
> > > ________________________________________
> > > "Why not all convex combinations of two, or possibly more, samples?
> > > -------------------------------------------------------
> > > All convex combinations of two samples can be sampled (some with a lower frequency than others). We also considered convex combinations of more samples but decided that in the version of GAIA we would use only two.  We mention sampling convex combinations from more than two samples (e.g. convex combinations of the entire batch)  in the conclusion but decided leave that up to future work because our computational resources were not large enough to use large batch sizes for the bigger datasets (e.g. CELEB-A).  We would be interested in pursuing this question in future work but decided that exploring other interpolations was beyond the scope of the current paper.
> > >
> > > ________________________________________
> > > "To conclude, the paper presents quite good qualitative results on the CelebA-HQ dataset, but has problems with the thoroughness of the experimental evaluation, discussion of the related work, and presentation. The paper cannot be published in its current form.
> > > -------------------------------------------------------
> > > We hope that our revisions have met the standards for publication in ICLR by this reviewer, and thank the reviewer for their time, expertise, and effort in improving our publication.

---

### Public Comment · (anonymous) · 2018-09-30
**Comparison to prior work with special latent distributions?**

How does your method compare to the work in Semantic Interpolation in Latent Models ( https://arxiv.org/pdf/1710.11381.pdf ), which instead used a variant of the gamma distribution as their latent prior to encourage the latent distribution to suit linear semantic interpolation? (The choice of distribution ensures that the distribution of midpoints between two randomly sampled latent points has the same form as the distribution of latent points, with a limited constant divergence even at high dimensionalities - it avoids the "normal distributions in high dimensions are thin-shelled bubbles" problem.) While obviously your method is more powerful and flexible than the simple choice of a slightly different prior, Kilcher et al. do seem to show improvements in linearity and convexity of their latent distribution as well (with the image interpolation examples showing that the center of the distribution seems to correspond to a "generic" image rather than a low-density blur, and rather dramatic visual improvements for straight Euclidean interpolation as well as quantitative improvements in the semantic-algebra "king - man + woman = queen" domain.)

It would be interesting to compare how much the more complex method in your work buys you. (I suspect it'd be substantial, of course - but it would address the question of how much of your improvement is because of the added flexibility for general-purpose distribution-matching, and how much is simply because you used *anything* smarter than a normal distribution)

---

> ### Public Comment · (anonymous) · 2018-10-15
> **re: Comparison to prior work with special latent distributions?**
>
> That's a great question/idea for comparison. Currently the only GAN architectures that we have compared are those with an inference method, because for our practical purpose (stimuli generation in behavioral and electrophysiology experiments) we are interpolating between real stimuli (e.g. animal vocalizations) for playback. Of course, an inference method could be added to Kilcher et al., in several of the ways used in the papers we cited.
>
> With the right dataset, I think we could come up with a better comparison method between models for learning convex or more linear latent space interpolations. For example, if you look at work from Simoncelli's group (e.g. https://youtu.be/jahYVzvTtJw?t=1502 and http://www.cns.nyu.edu/pub/lcv/henaff16b-reprint.pdf) they performed linear transformations on images (and also just took video) and look at (1) smoothness of interpolations between points in latent space, and (2) the linearity and curvature of real sequences projected in latent space. Training our network to interpolate between, for example, frames of video data and computing the curvature of sequences of intervening frames projected into latent space could be a good way to compare how these networks linearize latent representations.

---

### Public Comment · (anonymous) · 2018-10-30
**Some Questions and Related Work**

This paper propose an interesting neural network architecture combining AE and GAN and improving the convexity of latent space. I am confusing about how the generated process works. As the paper claims there is no prior constraint like VAE, then how do the network generate new samples? It seems that how to generate from latent space is not described in details.  From the results showed in the paper,  I want to know whether two real data samples is needed when generated a new sample.

Another relevant work on latent space interpolation is https://arxiv.org/pdf/1807.07543.pdf, which is focused on the latent interpolation of VAE. And the Adversarially Constrained Autoencoder Interpolation (ACAI) regularizer proposed in their paper is similar to the idea of this paper. I suggest more comparison should be conducted on the related works.

---

> ### Author Response · Authors · 2018-10-30
> **Thank you for your questions**
>
> >"Another relevant work on latent space interpolation is https://arxiv.org/pdf/1807.07543.pdf, which is focused on the latent interpolation of VAE. And the Adversarially Constrained Autoencoder Interpolation (ACAI) regularizer proposed in their paper is similar to the idea of this paper. I suggest more comparison should be conducted on the related works. "
>
> We recognize the similarities between GAIA (ours) and ACAI. Because ACAI was not yet published when we published our paper on arXiv, it was not included in our review of similar works (it was posted around the same time, just a few days later). That said, we have discussed the similarities between these works with the authors of ACAI. The primary difference between the two implementations is a difference in the adversarial regularizer used - we used an autoencoder as discriminator and they used a traditional discriminator. I agree that it would be interesting to compare the two methods.
>
> > "...which is focused on the latent interpolation of VAE."
>
> We should note that neither ACAI nor GAIA (ours) involves latent interpolation of a VAE or variational regularization. However in the ACAI paper they provide several comparisons with different forms of VAEs.
>
> > "I am confusing about how the generated process works. As the paper claims there is no prior constraint like VAE, then how do the network generate new samples? It seems that how to generate from latent space is not described in details. From the results showed in the paper,  I want to know whether two real data samples is needed when generated a new sample."
>
> You are correct that there is no prior constraint on GAIA (we do constrain the latent space using the local-structure loss from 2.1 however). Two real data samples are not needed to generate a new sample (any convex combination of samples can be projected into the network). However, there is no explicit training to ensure that convex combination lies on the data manifold (see Figure 1D). In fact, if you look at the left half of Figure 4, these attribute vectors are convex combinations of all data points with each attribute, and do not look realistic.

---

### Meta-Review · Area_Chair1 · 2018-12-14
**Interesting idea but insufficient evaluation of the method to establish benefits over existing methods**

**Confidence:** 5
**Recommendation:** Reject

**Metareview:**

The idea of the paper -- imposing a GAN type loss on the latent interpolations of an autoencoder -- is interesting. However there are strong concerns from R2 and R3 about limited experimental evaluation of the proposed method which falls short of demonstrating its advantages over latent spaces learned by existing GANs. Another point of concern was the use of only one real dataset (CelebA). Authors made substantial revisions to the paper in addressing many of the reviewers' points but these core concerns still persist with the current draft and it's not ready for publication at ICLR. Authors are encouraged to address these concerns and resubmit to another venue.